# How Do Football Playing Positions Differ in Body Composition? A First Insight into White Italian Serie A and Serie B Players

**DOI:** 10.3390/jfmk8020080

**Published:** 2023-06-15

**Authors:** Tindaro Bongiovanni, Alessio Rossi, Federico Genovesi, Giulia Martera, Giuseppe Puleo, Carmine Orlandi, Mirco Spedicato, F. Marcello Iaia, Riccardo Del Vescovo, Stefano Gallo, Roberto Cannataro, Patrizio Ripari, Matteo Levi Micheli, Stefania Cataldi, Athos Trecroci

**Affiliations:** 1Department of Biomedical and Neuromotor Sciences, University of Bologna, 40126 Bologna, Italy; tindaro.bongiovanni2@unibo.it; 2Department of Performance, Palermo Football Club, 90146 Palermo, Italy; g.puleo@palermofc.com; 3Department of Computer Science, University of Pisa, 56126 Pisa, Italy; 4National Research Council (CNR), Institute of Information Science and Technologies (ISTI), 56124 Pisa, Italy; 5Medical Department, Manchester City Football Club, Manchester M11 3FF, UK; federico.genovesi@mancity.com; 6Department of Performance Nutrition, Spezia Calcio, 19123 La Spezia, Italy; 7Department of Sport Science, Tor Vergata University of Roma, 00133 Roma, Italy; carmine.orlandi@uniroma2.it; 8Department of Nutrition, U.S. Lecce Football Club, 73100 Lecce, Italy; info@mircospedicato.it; 9Department of Biomedical Science for Health, University of Milan, 20133 Milan, Italy; marcello.iaia@unimi.it (F.M.I.); athos.trecroci@unimi.it (A.T.); 10Department of Performance, Hellas Verona Football Club, 37135 Verona, Italy; delvescovoriccardo@gmail.com (R.D.V.); stefanogallonutrizionista@gmail.com (S.G.); 11Villa Stuart Clinic, FIFA Medical Center of Excellence, 00186 Rome, Italy; 12Department of Pharmacy, Health and Nutritional Sciences, University of Calabria, 87036 Rende, Italy; r.cannataro@gmail.com; 13Department of Technological Medicine, University of Chieti-Pescara, 66100 Pescara, Italy; patrizio.ripari@unich.it; 14Department of Experimental and Clinical Medicine, University of Florence, 50100 Florence, Italy; matteo.levimicheli@unifi.it; 15Department of Translational Biomedicine and Neuroscience (DiBraiN), University of Study of Bari, 70126 Bari, Italy; stefania.cataldi@uniba.it

**Keywords:** anthropometry, soccer, morphology, bioelectrical impedance analysis

## Abstract

The present study aimed to investigate how playing positions differ in specific body composition variables in professional soccer players with respect to specific field zones and tactical lines. Five hundred and six Serie A and B professional soccer players were included in the study and analyzed according to their playing positions: goalkeepers (GKs), central backs (CBs), fullbacks (FBs), central midfielders (MIDs), wide midfielders (WMs), attacking midfielders (AMs), second strikers (SSs), external strikers (ESs), and central forwards (CFs), as well as their field zones (central and external) and tactical lines (defensive, middle, and offensive). Anthropometrics (stature and body mass) of each player were recorded. Then, body composition was obtained by means of bioelectric impedance analysis (BIA). GKs and CFs were the tallest and heaviest players, with no differences from each other. Likewise, GKs and CFs, along with CBs, were apparently more muscular (for both upper and lower limbs) and fatter at the same time compared with the other roles. Overall, players of the defensive line (CBs and FBs), along with those playing in central field zones (CBs, MIDs, AMs, SSs, and CFs), were significantly (*p* < 0.05) superior in almost all anthropometric and body composition variables than those of middle and offensive line and external zones, respectively.

## 1. Introduction

Soccer is an intermittent team sport requiring players with well-developed physical, psychological, tactical and technical skills [1]. From a physiological perspective, soccer’s demands are complex and vary depending on different factors such as players’ performance level for both teammates and opponents, the style of play adopted by the two contending teams, and positional roles that imply specific demands [2,3,4]. For example, the goalkeepers (GKs) cover approximately 50% less total distance and <10% of the distance at high-intensity speed (>19.8 km/h) than outfield players [5,6], also including very brief explosive movements [7] in their locomotive match demands. Central midfielders (MIDs) cover more total and high-intensity running distances than center backs (CBs), while wide midfielders (WMs) sprint over greater distances than defensive linemen [8]. Further, both full backs (FBs) and MIDs exhibit a higher number of ball possessions compared to other positions [9] that would further differentiate their physical and technical performance on the pitch. Research demonstrates that MIDs and FBs display the highest VO2 max and show the greatest physical capacities by means of exhaustive intermittent running tests [10].

Based on the aforementioned match activity profiles, it stands to reason that players’ anthropometry and body composition profiles may be differently characterized as a function of their playing position [4]. A previous study showed that GKs and central forwards (CFs) were taller, heavier, and fatter with respect to MIDs [4]. Similarly, Anderson et al. (2019) [7] underreported higher skinfold thicknesses, as a proxy marker of body fat, in GKs compared with outfield players. However, Mala et al. (2017) [11] did not find any differences among playing positions in body fat. However, they observed a significantly higher lean body mass in GKs compared to defensive (FBs, CBs), middle (WMs and MIDs), and offensive (CFs) linemen. Other authors found that professional GKs were leaner than FBs and MIDs [12]. The fact that GKs are leaner than some outfield players might be the result of regularly engaging in additional resistance training to cope with the increasing demands of modern soccer [7]. Indeed, modern soccer requires GKs to execute fast and explosive actions such as changing directions, jumping, and diving [7], while continuously shifting their center of gravity within a larger operating area that is closer to outfield players [11]. 

Thus, retrieving specific data on the estimated body composition becomes relevant in a real-world setting, as this could provide practitioners with a deep insight into their athletes’ potential while tracking training-related effects. However, limited data are available to describe physical differences linked to playing position. Moreover, scant evidence also exists on how players belonging to different tactical lines (defensive, middle, and offensive) and field zones (central and external) differ in body composition.

Therefore, the aim of this study was to investigate how playing positions differ in specific body composition variables at an individual level. Secondly, this study aimed to evaluate potential differences in the same selected body composition variables by field lines and zones.

## 2. Materials and Methods

### 2.1. Subject

A total of 506 elite white soccer players (age = 25.72 ± 4.02 years, height = 183.72 ± 5.86 cm, body mass = 79.91 ± 6.31 kg) competing in professional Italian soccer teams (Serie A and Serie B) voluntarily took part in this study. Players were separated into nine playing positions according to the role typically attributed by their coaching staff (*n* = 57 goalkeepers—GK, *n* = 82 central backs—CB, *n* = 73 fullbacks—FB, *n* = 43 central midfielders—MID, *n* = 66 wide midfielders—WM, *n* = 42 attacking midfielders—AM, *n* = 37 s strikers—SS, *n* = 52 external strikers—ES, and *n* = 54 central forwards—CF) in accordance with the zone of the pitch they used to play. All the participants signed their informed consent before taking part in the experiment (Ethics committee, University of Milan, approval code: 32/16 of 16 November 2016), which complies with the principles of the Declaration of Helsinki.

### 2.2. Data Acquisition

#### 2.2.1. Procedure

This quantitative study involved the participation of 12 elite soccer clubs, and the assessment was made for each team during the in-season period (in October). Players’ body compositions were recorded in the morning (from 8.30 a.m. to 9.30 a.m.) following a 12 h food and fluid fast. All participants were also asked to abstain from drinking caffeinated and alcoholic beverages within 24 h before testing. Likewise, the participants were asked to avoid vigorous exercise within 24 h before assessing, as in previous studies [13].

#### 2.2.2. Body Composition

Body mass and height (or stature) were measured to the nearest 0.1 kg and 0.1 cm, respectively, via a portable stadiometer (Seca 213, Hamburg, Germany) and a flat scale (Seca 877, Hamburg, Germany) while barefoot and wearing a bathing suit. 

Bioelectrical impedance analysis (BIA) was performed using a phase-sensitive bioelectrical analyzer (BIA 101 BIVA PRO, Akern, Florence, Italy). The device emits an alternating sinusoidal electric current of 250 μA at an operating monofrequency of 50 kHz (±0.1%). The device was calibrated using the standard control circuit supplied by the manufacturer that has a known impedance. Participants were positioned supine with a leg opening of 45° with respect to the midline of the body, and with the upper limbs positioned 30° away from the trunk. After cleaning the skin with alcohol pads, four adhesive electrodes (Biatrodes Akern Srl, Florence, Italy) were placed on the backs of the hands and another four electrodes on the ankles of the corresponding feet, keeping 5 cm between each electrode [14]. The proximal hand electrode was positioned between the radial and ulnar styloid processes, directly superficial to the distal radioulnar joint. The distal hand electrode was positioned in the center of the third, proximal phalanx. The proximal foot electrode was placed directly between the medial and lateral malleoli at the ankle. The distal foot electrode was placed immediately proximal to the second and third metatarsophalangeal joints [15]. Resistance (Rz), reactance (Xc), and phase angle (PhA) raw data were obtained. R is the opposition to the flow of an injected alternating current, at any current frequency, through intra and extracellular ionic solutions, while Xc represents the dielectric or capacitive component of cell membranes, organelles, and tissue interfaces [16]. From the raw BIA variables, estimates of appendicular arm lean soft tissue (ALST) and leg lean soft tissue (LLST) were obtained as previously [15]. Fat-free mass (FFM) was estimated using the athlete-specific equation of Matias et al. [17] and, consequently, fat mass (FM) was derived by subtracting the body mass minus the fat-free mass in kilograms. 

As regards PhA, it has been suggested as a biomarker of cellular health and cell membrane integrity and descriptive of the intracellular (ICW)–extracellular (ECW) water ratio [15].

In addition, estimates of total body water (TBW) and ECW were obtained using equations (Equations (1) and (2)) specific for athletes by Matias et al. [17], where sex is a binary value where 0 and 1 refer to female and male, respectively.
TBW (L) = 0.286 + 0.195 × stature2/Rz + 0.385 × body mass + 5.086 × sex(1)
ECW (L): 1.579 + 0.055 × stature2/Rz + 0.127 × body mass + 0.006 + stature2/Xc + 0.932 × sex(2)

ICW was then calculated by subtracting ECW from TBW.

### 2.3. Statistical Analysis

One-way analysis of variance (ANOVA) was performed in order to detect statistical differences among playing positions, field lines (defensive, middle, and offensive), and zones (central or external position) for each selected variable. The normality of data distribution assumption was assessed by Shapiro–Wilks’ normality test. Additionally, Tukey’ post hoc pairwise comparison analysis was performed when ANOVA showed statistical significance. All the analyses were performed in the Python 3.8 programming language. The statistical significance was set at 0.05 (5%).

## 3. Results

Table 1 shows the descriptive statistics of all the playing positions. Several statistically significant differences among playing positions were detected. From a descriptive point of view, GKs were the tallest and heaviest players, also exhibiting the highest values in TBW, ECW, ICW, FFM, FM, ALST, and LLST. GKs also presented the lowest PhA values along with AMs, while MIDs had the highest. ESs were the lightest players, also exhibiting the lowest values in TBW, ECW, ICW, FFM, ALST, and LLST. FBs were the leanest players (Figure 1a). The post hoc outputs linked to statistical significance are included in the Appendix A for height (Appendix A), weight (Appendix A), PhA (Appendix A), TBW (Appendix A), ECW (Appendix A), ICW (Appendix A), FFM (Appendix A), FM (Appendix A), ALST (Appendix A), and LLST (Appendix A). Specifically, the post hoc analyses revealed that GKs were significantly different (*p* < 0.01) in stature from CBs, MIDs, WMs, AMs, SSs, and ESs. CBs were also significantly different (*p* < 0.01) in stature from FBs, MIDs, AMs, SSs, and ESs. FBs were also significantly different (*p* < 0.01) from CFs. MIDs were also significantly different (*p* < 0.01) in stature from WMs and ESs, and CFs. WMs were also significantly different (*p* < 0.01) in stature from SSs, ESs, and CFs. AMs were also significantly different (*p* < 0.01) in stature from SSs and CFs. Moreover, ESs were also significantly different (*p* < 0.01) in stature from CFs.

As regards body mass, GKs was significantly different (*p* < 0.01) in body mass from MIDs, WMs, AMs, SSs, and ESs. CBs were significantly different (*p* < 0.01) in body mass from FBs, MIDs, WMs, AMs, SSs, and ESs. FBs, MIDs, WMs, and AMs were also significantly different (*p* < 0.01) in body mass from ESs and CFs. ESs were also significantly different (*p* < 0.01) in body mass from CFs.

For a better visual inspection of each field position, the post hoc analyses of the selected variables (PhA, ECW, ICW, FFM, FM, ALST, and LLST) are represented in Figure 2 by pitches in the form of tactical systems.

Table 2 and Table 3 report the descriptive statistics of the players belonging to specific field lines (defensive, middle, and offensive) and zones (central and external). Except for PhA and fat mass (FM), players of the defensive line were significantly (*p* < 0.05) superior in all variables compared with middle and offensive linemen. Likewise, players of the central zone of the pitch (CBs, MIDs, SSs, and CFs) were superior (*p* < 0.05) in all variables than external players except for PhA.

## 4. Discussion

The main findings of this study revealed that, along with anthropometry, different body composition profiles are identified by playing position in professional soccer players. From the current analysis, it emerged that GKs and CFs were the tallest and heaviest players, with no differences from each other. Specifically, GKs possess their own distinctive anthropometric and body composition characteristics in terms of ECW, ICW, FFM, FM, ALST, and LLST, especially compared to FB, MID, AM, WM, SS and ES. These characteristics are also largely shared by CFs.

GKs and CFs exhibited the highest values in both ALST and LLST along with FFM, while SS and ES presented the lowest. ALST and LLST are derived measures of skeletal muscle, identifying the largest non-adipose tissue component of an individual’s body composition (Quinterio et al., 2009) [18]. Compared with outfield players, GKs regularly engage in additional resistance training for maintenance and growth of muscle mass [7], which would justify the higher ALST and LLST values. In turn, aside from performing the most high to very high intensity activity, CFs undergo the most contact situations, imposing pushing and pulling activities for both the upper and lower body [19]. It is likely that their specific need to be physically prepared would lead them to emphasize additional strength-related training compared with other field-based positions. This result supports the consistent link between the appendicular lean soft tissue of both the upper and lower body [20] in obtaining informative data on regional muscular mass.

If, on the one hand, GKs seem to exhibit much greater muscle mass in both upper and lower limbs, on the other hand, they are fatter [21,22] than other outfield players (e.g., FB, MID, ES) along with CBs and CFs. To the best extent of the authors’ knowledge, this represents a novelty within the literature that should be investigated in depth. Similarly, Mala et al. (2017) [11] reported the highest values for FFM and FM in under-19 elite GKs, even though significant differences were observed only for FFM. Routinely, the weekly training loads accumulated by the outfield players are greater than GKs to influence their energy expenditure [7], and consequently, their percentage of fat. Separate discussion for CBs and CFs, whose FM levels might depend on other factors (e.g., specific duties on the pitch). For instance, extra mass, albeit inactive as fat tissue, may be an advantage in hand-to-hand actions commonly experienced by both CFs and CBs. Indeed, CFs often have to hold the ball and shield it while dueling with defensive linemen (e.g., CB) that seek to win the ball. Conversely, FBs were the leanest players, supporting their dynamically demanding role on the pitch.

ECW and ICW data represent additional sources of information to control for potential body composition changes closely linked to players’ playing position and their on-field performance [23]. Enhanced cellular hydration (i.e., ICW increases) may be indicative of increased glycogen synthesis (because of the highly osmotic features of glycogen) that would promote anabolism via cellular swelling [24,25]. At this point, players might benefit from this condition from a muscular function point of view. In fact, ICW was previously observed as one of the best predictors for jumping height performance in male professional soccer players [23]. According to the present findings, between-role differences of both ECW and ICW collectively matched those of FFM, especially for GKs, CBs, and CFs, who reported the highest values. This result seems to reflect the anabolic adjustment via cellular swelling capable of stimulating pathways that could increase protein synthesis. As a consequence, muscular players would also exhibit high ICW and ECW, which is grounded in earlier associations of body water with upper-body strength levels in individual [26,27] and team sport athletes [28].

PhA was highest for MIDs compared with AMs and GKs, who had the lowest values. An athlete with a higher PhA value has a greater muscle mass and a higher cellular integrity [29], putting him/her at a greater advantage during explosive action [20]. However, results of ALST and LLST by GKs appear not supporting this. Of note, it is worth noting that PhA is also considered a prognostic marker of cell health due to its positive effect on physical activity [30]. The process through which physical activity acts on PhA appears to entail a variety of mechanisms, which manifest in a better integrity and functionality of the cell membrane, changes in intracellular composition, and enhanced tissue capacity [31]. If transferred into the real-world setting, it might be assumed that PhA can discriminate between players’ physical activity profiles. In keeping, a moderate association between PhA and short-term maximal intensity efforts in soccer players [32] was found. For instance, this would reflect the match demand activities of MIDs versus GKs. Unfortunately, at present, this remains speculative due to the lack of evidence. Further research will have to establish whether the role of match-based physical activities linked to specific playing position is explained by PhA outcomes. This information would be relevant for the coaching staff when arranging ad hoc monitoring processes over the season.

An interesting side finding of this study was that players of middle and offensive lines presented different body composition profiles from defensive linemen, who were higher in stature, body mass, and indirect measures of muscle mass (e.g., ALST, LLST, FFM, ECW, ICW). It should be noted that the defensive line consists of FBs and CBs, who presented anthropometric profiles at odds. Although FBs were the leanest, the anthropometric and body composition characteristics of CBs made a substantial upward contribution in the differentiation against middle (e.g., MID) and offensive linemen (e.g., CF). Yet, although CFs present similar characteristics, ESs’ data made a substantial downward contribution that provided outcomes for offensive linemen comparable with middle linemen. Moreover, outfield players competing within the central zones (e.g., CB, MID, AM, and CF) were taller, heavier, fatter, and more muscled than those of the external zones (i.e., FB, WM, ES, and SS). A likely explanation may be attributed to the fact that players in the central zones (in and outside the box or along the midfield) of the pitch perform within a crowded area in which collision, hand-to-hand duels, and tackles are on the agenda during a match. Altogether, these actions require specific anthropometric and body composition profiles that can be easily identifiable by the current results.

This study is not devoid of limitations. Potential factors that could influence body composition profiles, such as dietary habits, individual training regimens, or injury history, were not controlled. Future research should account for these additional factors to provide a more comprehensive understanding of the current findings. Furthermore, BIA equations are typically developed and validated on specific populations, often with limited diversity in terms of age, sex, ethnicity, and body composition characteristics. Thus, our results are exclusive to white players. This would introduce a potential ethnic bias and restricts the generalizability of our findings to individuals from other ethnic backgrounds.

## 5. Conclusions

This study disclosed different anthropometric and body composition profiles among playing positions in professional soccer players. In particular, the present findings showed that GKs and CFs were the tallest and heaviest players, with no differences from each other. Additionally, GKs and CFs, along with CBs, showed greater body muscularity and fatness at the same time, as opposed to the other roles. Of note, playing in the central zone of the pitch makes ball possession difficult due to the high density of players and the numerous hand-to-hand contacts. This supports the current side findings, in which players competing in the central zones (CB, MID, AM, SS, and CF) of the pitch and in the defensive line (CB and FB) presented the highest stature and body mass as well as the highest measures of muscle mass (e.g., ALST and LLST) and body fat (FM), increasing their efficacy during duels. Taken together, these findings may be of relevance for designing position-specific training programs that would help, for example, GKs and CFs to exploit their high level of muscle mass by focusing on additional strength- and power-based stimuli.

## Figures and Tables

**Figure 1 jfmk-08-00080-f001:**
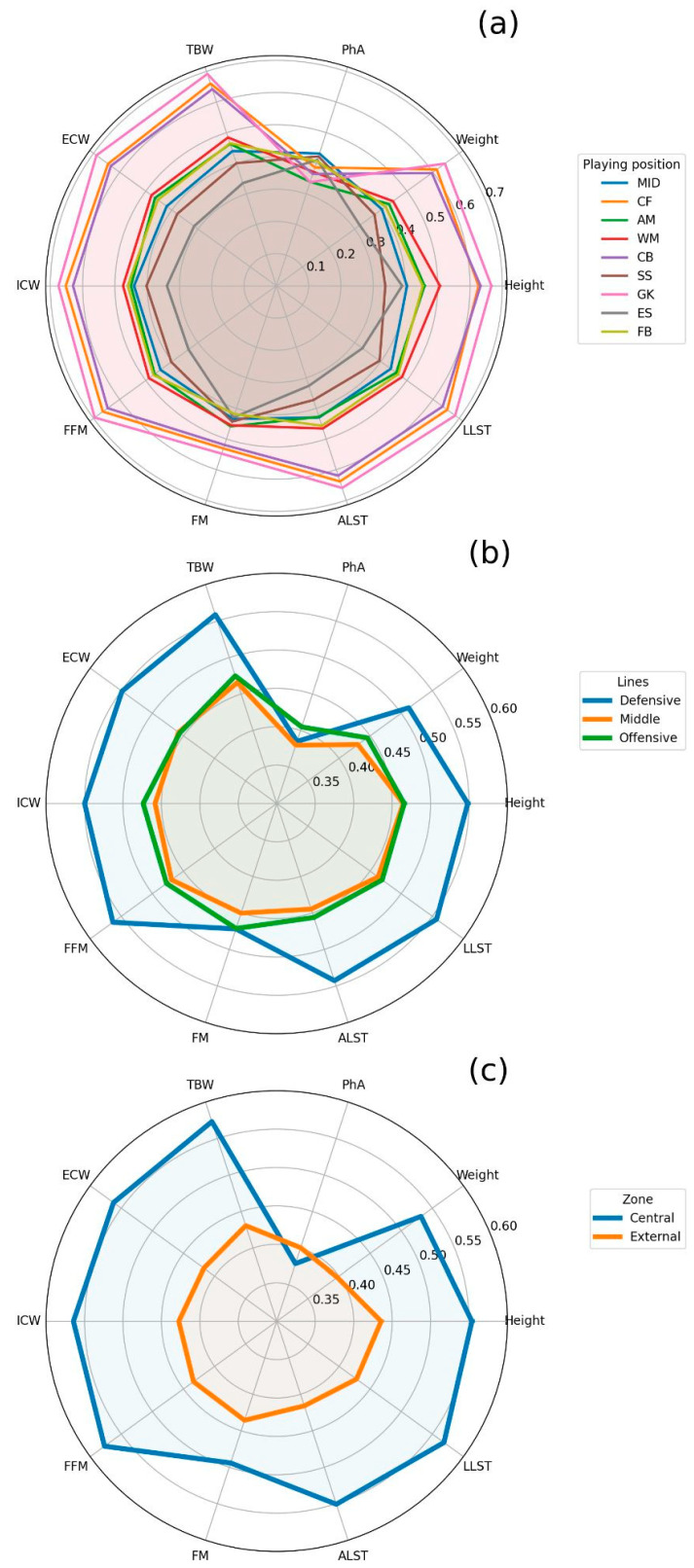
Radar chart with the normalized variables per playing position. (**a**) refers to playing position, (**b**) shows the playing lines, and (**c**) is linked to the pitch zone. Note: PhA = phase angle, TBW = total body water, ECW = extracellular water, ICW = intracellular water, FFM = fat-free mass, FM = fat mass, ALST = arm lean soft tissue, LLST = leg lean soft tissue, GK = goalkeeper, CB = central back, FB = fullback, MID = central midfielder, WM = wide midfielder, AM = attacking midfielder, SS = second striker, ES = external striker, CF = central forward. The features were normalized by min-max standard scaler on the entire dataset. The maximum value refers to 1, while the minimal one to 0.

**Figure 2 jfmk-08-00080-f002:**
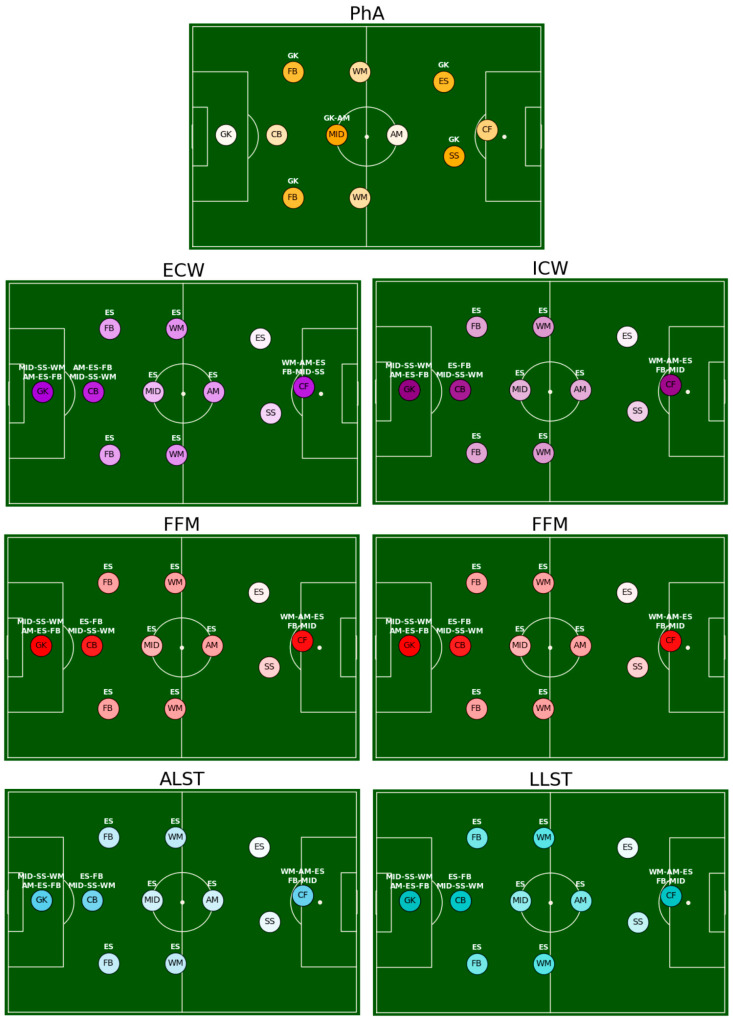
Differences between players’ playing positions for PhA (phase angle), ECW (extracellular water), ICW (intracellular water), FFM (fat-free mass), FM (fat mass), ALST (arm lean soft tissue), LLST (leg lean soft tissue). Values increase with darker colors. GK = goalkeeper, CB = central back, FB = fullback, MID = central midfielder, WM = wide midfielder, AM = attacking midfielder, SS = second striker, ES = external strikers, CF = central forward. The darker the color of the dots is, the higher the values of the specific playing position is.

**Table 1 jfmk-08-00080-t001:** Descriptive statistics expressed as mean (± standard deviation) and statistical analysis. ANOVA statistical significance between playing position: * *p*-value < 0.001.

Features	GK	CB	FB	MID	WM	AM	SS	ES	CF	ALL
Height (cm) *	189.15 (3.50)	187.96 (3.51)	181.61 (4.03)	179.87 (4.81)	183.49 (4.58)	181.80 (4.78)	177.49 (5.08)	179.34 (5.42)	187.74 (4.73)	183.72 (5.86)
Body mass (kg) *	85.89 (4.66)	84.04 (4.11)	77.48 (4.31)	76.96 (5.03)	78.54 (5.58)	77.98 (5.32)	75.92 (4.40)	73.83 (5.07)	84.73 (5.54)	79.91 (6.31)
PhA (°) *	7.83 (0.60)	7.96 (0.65)	8.18 (0.58)	8.30 (0.78)	8.00 (0.55)	7.83 (0.61)	8.25 (0.72)	8.20 (0.37)	8.07 (0.64)	8.06 (0.63)
TBW (L) *	53.85 (2.73)	52.88 (2.41)	49.42 (2.56)	48.93 (2.83)	49.79 (3.27)	49.38 (2.67)	48.15 (2.22)	46.88 (3.08)	53.24 (3.03)	50.54 (3.59)
ECW (L) *	21.23 (1.13)	20.81 (0.97)	19.44 (1.03)	19.18 (1.17)	19.62 (1.31)	19.52 (1.06)	18.87 (0.77)	18.38 (1.22)	20.89 (1.10)	19.87 (1.43)
ICW (L) *	32.62 (1.66)	32.07 (1.51)	29.99 (1.56)	29.75 (1.73)	30.17 (1.99)	29.87 (1.66)	29.28 (1.49)	28.50 (1.88)	32.35 (1.96)	30.66 (2.20)
FFM (kg) *	74.14 (4.06)	72.73 (3.63)	67.73 (3.82)	66.96 (4.22)	68.22 (4.83)	67.60 (3.86)	65.78 (3.22)	63.90 (4.61)	73.25 (4.47)	69.3 (5.29)
FM (kg) *	11.74 (2.08)	11.31 (2.28)	9.75 (1.80)	9.99 (2.51)	10.32 (1.62)	10.38 (2.00)	10.13 (1.76)	9.92 (2.24)	11.48 (2.52)	10.60 (2.21)
FM (%)	13.65 (2.18)	13.42 (2.51)	12.57 (2.05)	12.93 (2.83)	13.12 (1.71)	13.23 (1.92)	13.29 (1.79)	13.41 (2.75)	13.49 (2.55)	13.22 (2.29)
ALST (kg) *	7.51 (0.58)	7.35 (0.57)	6.72 (0.55)	6.61 (0.60)	6.75 (0.64)	6.61 (0.48)	6.39 (0.49)	6.21 (0.66)	7.43 (0.67)	6.90 (0.73)
LLST (kg) *	22.12 (1.37)	21.66 (1.24)	20.05 (1.29)	19.78 (1.43)	20.19 (1.61)	19.97 (1.25)	19.37 (1.05)	18.74 (1.57)	21.83 (1.49)	20.54 (1.76)

Note: PhA = phase angle, TBW = total body water, ECW = extracellular water, ICW = intracellular water, FFM = fat-free mass, FM = fat mass, ALST = arm lean soft tissue, LLST = leg lean soft tissue, GK = goalkeeper, CB = central back, FB = fullback, MID = central midfielder, WM = wide midfielder, AM = attacking midfielder, SS = second striker, ES = external strikers, CF = central forward.

**Table 2 jfmk-08-00080-t002:** Descriptive statistics expressed as mean (standard deviation) and statistical analysis regarding the tactical lines. Statistical (*p* < 0.05) difference: ^a^ and ^b^ difference from middle and offensive.

Features	Defensive	Middle	Offensive
Height (cm) ^a,b^	184.97 (4.92)	181.99 (4.91)	182.03 (6.77)
Body mass (kg) ^a,b^	80.95 (5.33)	77.93 (5.36)	78.49 (7.08)
PhA (°)	8.06 (0.63)	8.04 (0.66)	8.16 (0.59)
TBW (L) ^a,b^	51.25 (3.02)	49.43 (2.99)	49.61 (4.05)
ECW (L) ^a,b^	20.16 (1.21)	19.47 (1.21)	19.45 (1.56)
ICW (L) ^a,b^	31.09 (1.85)	29.97 (1.83)	30.16 (2.51)
FFM (Kg) ^a,b^	70.38 (4.47)	67.69 (4.41)	67.92 (5.97)
FM (kg)	10.57 (2.20)	10.24 (2.00)	10.57 (2.34)
ALST (Kg) ^a,b^	7.05 (0.64)	6.67 (0.59)	6.72 (0.83)
LLST (Kg) ^a,b^	20.9 (1.50)	20.01 (1.46)	20.07 (1.99)

Note: PhA = phase angle, TBW = total body water, ECW = extracellular water, ICW = intracellular water, FFM = fat-free mass, FM = fat mass, ALST = arm lean soft tissue, LLST = leg lean soft tissue.

**Table 3 jfmk-08-00080-t003:** Descriptive statistics expressed as mean (standard deviation) and statistical analysis regarding field zones.

Features	Central	External
Height (cm) *	185.16 (5.55)	180.97 (5.14)
Body mass (kg) *	81.68 (5.92)	76.70 (5.18)
PhA (°)	8.03 (0.68)	8.14 (0.56)
TBW (L) *	51.54 (3.29)	48.74 (3.06)
ECW (L) *	20.27 (1.28)	19.16 (1.23)
ICW (L) *	31.27 (2.05)	29.59 (1.87)
FFM (Kg) *	70.76 (4.85)	66.68 (4.54)
FM (kg) *	10.92 (2.40)	10.02 (1.86)
ALST (Kg) *	7.08 (0.69)	6.56 (0.63)
LLST (Kg) *	21.02 (1.62)	19.68 (1.53)

* Statistical (*p* < 0.05) difference. Note: PhA = phase angle, TBW = total body water, ECW = extracellular water, ICW = intracellular water, FFM = fat-free mass, FM = fat mass, ALST = arm lean soft tissue, LLST = leg lean soft tissue.

## Data Availability

All the relevant data are provided in the text and Appendix A.

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
