# Peer review of "How Do Football Playing Positions Differ in Body Composition? A First Insight into White Italian Serie A and Serie B Players"

_jfmk, 2023, doi:10.3390/jfmk8020080_

Round 1

Reviewer 1 Report

This is a study that reported differences in body size and body composition of elite soccer players using bioelectrical impedance analysis (BIA) technique. The authors reported greater body size and muscular among defensive players compared with players of other positions.

While the study is based on a relatively large sample size of elite soccer players, the authors failed to show if there are any differences or variabilities in body size and body composition between ethnic backgrounds of participants. In addition, since the study included participants from both Serie A and Serie B, there is a possibility that physique of the participants may differ depending on the level of the league they play. Therefore, it is suggested the authors to consider ethnicity and the level of the league they play in the analysis before they put all participants into one group.

In addition, it is not clear how the authors standardized level of each variable in Figure 1 as each variable is different in their unit. The authors are required to describe method of standardizing each variable to allow radar chart in the methods section. Furthermore, since the participants are assumed to be multi-ethnic population, it is uncertain if the prediction equations for TBW and ECW utilized in the study to be valid. Lastly, the authors failed to state limitations of the study. In order to clarify validity of the methodology, the authors are suggested to provide sufficient description with references.

Author Response

This is a study that reported differences in body size and body composition of elite soccer players using bioelectrical impedance analysis (BIA) technique. The authors reported greater body size and muscular among defensive players compared with players of other positions.

While the study is based on a relatively large sample size of elite soccer players, the authors failed to show if there are any differences or variabilities in body size and body composition between ethnic backgrounds of participants. In addition, since the study included participants from both Serie A and Serie B, there is a possibility that physique of the participants may differ depending on the level of the league they play. Therefore, it is suggested the authors to consider ethnicity and the level of the league they play in the analysis before they put all participants into one group.

Authors’ response: Thank you for the suggestion. Firstly, to reduce the inter-subject variability we have included only Caucasian people. We apologize for the confusion. We did not add this information in the first submission. Now, we have specified it within the title and and method section. Secondly, even though a statistical difference was detected among players competing in the two leagues, we kept all players together since the level of the competition was not an important features discriminating playing positons. It was used as independent features during clustering, resulting useless. Splitting the players into two groups accordingly with the level of competition would not increase the model accuracy.

In addition, it is not clear how the authors standardized level of each variable in Figure 1 as each variable is different in their unit. The authors are required to describe method of standardizing each variable to allow radar chart in the methods section.

Authors’ response: The features was normalized by min-max standard scaler on the entire dataset. The maximum values refer to 1, while the minimum equal to 0. We have added the description in the figure caption.

Furthermore, since the participants are assumed to be multi-ethnic population, it is uncertain if the prediction equations for TBW and ECW utilized in the study to be valid.

Authors’ response: To reduce the inter-subject variability we have included only Caucasian people. Please refer to the previous comment. Further details have been added within the title and method section.

Lastly, the authors failed to state limitations of the study.

Authors’ response: Thanks for the suggestion we have added a limitation paragraph at the end of the discussion section.

Reviewer 2 Report

Study question: Why was bioelectrical impedance used instead of more accurate testing method such as air displacement plethysmography? If the purpose was to use a testing method that would be more applicable in a coaching environment then please state so to defend it’s use.

Page 2, 2nd paragraph: These two sentences begin with the word “This” and not clear what is being references to. Please add the term/variable that is being referred to for clarity.

“This might be the result of modern soccer that requires a GK to execute fast and ex-plosive actions such as changing directions, jumping and diving [7], while continuously shifting the center of gravity within a larger operating area that is closer to outfield players [11]. This would also be the result of the current trend of GKs to regularly engage in additional resistance training in the attempts to cope with the increasing demands of modern soccer [7].”

Procedure Section: Change “observational” to “quantitative”.

Results section: The authors switch between body mass and weight. Pick one term. Mass is kg and weight is lbs.

Statistics section needs correct labeling.

Figure 2.: The letter is too small to read and the colors have no context. Increases by which magnitude, 10%; 20-30%, as this information may be critical on determining on the amount of strength & conditioning intervention is needed?

What were the limitations in this study? Please add a paragraph or two to address them in the discussion section.

Conclusion needs explanation on how a practitioner (e.g. sport coach, strength & conditioning coach, etc.) can apply this data to the athlete’s development.

Author Response

Study question: Why was bioelectrical impedance used instead of more accurate testing method such as air displacement plethysmography? If the purpose was to use a testing method that would be more applicable in a coaching environment then please state so to defend it’s use.

 Authors’ response: Thank you for such a comment as it allows us to express our point of view. We employed BIA due to its non-invasiveness, relatively low cost, and portability, favoring the athletes’ body composition and somatotype to be monitored with a good margin of accuracy (Bertuccioli et al. 2022). Additionally, all the clubs involved in the study have the same model of BIA. We are aware about the limitations on the use of BIA (i.e., isotropic assumption of the huma body, skin preparation, diuretic medications and so on). However, many studies on body composition and clinical condition evaluations have been carried out using this technique with promising results. On one hand, we are also aware that air displacement plethysmography (ADP) provides reliable results with minimal radiation exposure, being particularly useful for research studies in sports performance. On the other hand, ADP measurements may also still have inherent limitations and sources of error (i.e., assumption of a constant density of fat-free mass across individuals) that would limit its accuracy on large-scale.

Page 2, 2nd paragraph: These two sentences begin with the word “This” and not clear what is being references to. Please add the term/variable that is being referred to for clarity.

“This might be the result of modern soccer that requires a GK to execute fast and ex-plosive actions such as changing directions, jumping and diving [7], while continuously shifting the center of gravity within a larger operating area that is closer to outfield players [11]. This would also be the result of the current trend of GKs to regularly engage in additional resistance training in the attempts to cope with the increasing demands of modern soccer [7].”

Authors’ response: We have modified it accordingly.

Procedure Section: Change “observational” to “quantitative”.

Authors’ response: We would like to thank the reviewer for his/her suggestion. We have changed it.

Results section: The authors switch between body mass and weight. Pick one term. Mass is kg and weight is lbs.

Authors’ response: Thank you for the suggestion. We uses body mass instead of weight within the text.

Statistics section needs correct labeling.

Authors’ response: We would like to thank the reviewer for her/his suggestion. We have corrected it.

Figure 2.: The letter is too small to read and the colors have no context. Increases by which magnitude, 10%; 20-30%, as this information may be critical on determining on the amount of strength & conditioning intervention is needed?

Authors’ response: Thank you for the suggestion, we have adding the description of the color in the figure caption as: “The darker the color of the dots is, the higher the values of the specific playing position is.”. Moreover, the size of the letters was increased.

What were the limitations in this study? Please add a paragraph or two to address them in the discussion section.

Authors’ response: We have added a dedicated paragraph with the study limitation.

Conclusion needs explanation on how a practitioner (e.g. sport coach, strength & conditioning coach, etc.) can apply this data to the athlete’s development.

 Authors’ response: We have added a sentence referring to specific practical applications.